# Review on Ethnoveterinary Practices in Sub-Saharan Africa

**DOI:** 10.3390/vetsci8060099

**Published:** 2021-06-04

**Authors:** Ndahambelela Eiki, Nthabiseng Amenda Sebola, Bellonah Motshene Sakong, Monnye Mabelebele

**Affiliations:** Department of Agriculture and Animal Health, College of Agriculture and Environmental Sciences, University of South Africa, Florida, Roodepoort 1725, South Africa; sebolan@unisa.ac.za (N.A.S.); sakonbm@unisa.ac.za (B.M.S.); mabelm@unisa.ac.za (M.M.)

**Keywords:** ethnoveterinary, EVM, livestock health management, sub-Saharan Africa

## Abstract

**Background:** Livestock represents an important sector for the livelihood of sub-Saharan African countries’ inhabitants. In these countries, farmers raise livestock to meet household food demands and as additional sources of incomes, but its production is hampered by rampant animal diseases. The impact of animal diseases is particularly severe for poor communities that, although relying heavily on livestock, have limited access to modern veterinary services and therefore rely on indigenous medicines for the treatment of livestock ailments. **Methods:** The current review focuses on the ethnoveterinary health management practices found amongst livestock producers in sub-Saharan Africa. Documents were sourced from Google databases. **Results:** A total of 56 documents were reviewed, most of which were published recently (after 2000). The documents revealed the wide use of ethnoveterinary medicines among livestock producers in sub-Saharan African countries because of their cost and accessibility, threats to ethnomedicinal plant species through improper harvesting methods, overexploitation, the existence of inappropriate ethnoveterinary practices, and methods of conserving ethnomedicinal species. **Conclusions:** Given the persistent threats posed to ethnoveterinary medicine and/or practices in developing countries of sub-Saharan Africa, the findings of this review highlight the importance of integrating and promoting the use of ethnoveterinary medicine that is likely to be lost if it is not given enough importance. It is also important to obtain an overview of recent publications on ethnoveterinary medicines to identify the gaps and scope required to be filled by future studies. It is envisaged that the review will stimulate further ethnoveterinary research among livestock disease management practices, which could lead to new pharmaceuticals in the region.

## 1. Background

Livestock is vital for many economies in developing countries. Even though the livestock sub-sector contributes much to the national economies of sub-Saharan African countries, its development is hampered by different constraints [1]. These include rampant animal diseases, which result in loss of livestock and farm productivity, reduction in market opportunity, and impairment of human welfare. The impact of animal diseases is particularly severe for poor communities that, although relying heavily on livestock, have limited access to modern veterinary services.

McGaw and Eloff, [2] indicated that many livestock owners in rural areas are faced with relatively few veterinarians and shortages of other facilities, and traditional medicinal plants are the only choice to treat many ailments. Tolossa et al. [3] also reported that in many developing countries, the supply of modern veterinary drugs is constrained by scarcity, uncertain supply, and costs.

In view of these constraints, the search for alternatives becomes important, and sub-Saharan African stockowners are no exception, as these therapies are now widely used, for example, studies conducted in Angola [4], Botswana [5], Cameroon [6], Ethiopia [7], Kenya [8], Nigeria [9], Namibia [10], South Africa [11], Tanzania [12], Zambia [13], and Zimbabwe [14].

It is therefore imperative to respect, preserve, and maintain ethnoveterinary knowledge by documenting useful ethnoveterinary flora and conserving it for future generations. However, promoting the conservation and use of ethnoveterinary medicines does not mean downgrading or ignoring the value of modern medicine and attempting to replace one with the other but recognizing that both types have their strengths and limitations. In some instances, they complement each other; in other cases, ethnoveterinary practices will be a better choice, while in still others, modern practices should be recommended. Therefore, the purpose of this paper is to succinctly review the recent progress in ethnoveterinary medicines (EVM) and to create an understanding of ethnoveterinary medicines (EVM) by discussing the findings presented in recent research conducted in sub-Saharan Africa. In addition, it is important to review recent publications on EVM to find the gap and scope for further EVM research in the region.

## 2. Methods

The protocol of the current literature review was performed in agreement with the Preferred Reporting Items for Systematic Review (PRISMA-A) [15]. A comprehensive literature search was performed for studies written in English and concerning the use of ethnoveterinary medicines. The studies eligible in the qualitative synthesis (systematic review) were surveys that investigated the use of ethnoveterinary medicines in sub-Saharan Africa. In the search process, the keywords “ethnoveterinary,” “ethnobotany,” “ethnobotanical” “ethnomedicine,” “ethnomedicinal,” “medicinal plants,” “traditional healer,” “traditional medicine practitioner,” and “traditional medicine” were used. Studies were considered for inclusion if they dealt with ethnoveterinary medicine knowledge and ethnomedicinal plant species.

## 3. Results

### 3.1. Ethnoveterinary Medicinal Plants

Over the years, researchers have recorded medicinal plants used in treating animal diseases in sub-Saharan Africa. For example, Traore et al. [16] recorded 26 plants that are used to treat livestock diseases in Burkina Faso. Adeniran et al. [17] listed 31 medicinal plants used for the treatment of animal diseases, with the Fabaceae family constituting the highest proportion in the Federal Capital Territory, North-Central Nigeria. However, they contended that this may be due to the abundance of this plant family in their study area. Furthermore, their result is consistent with the findings of Ahmad et al. [18], who argued that the more common a plant taxon in an area, the greater the probability of its popular use. Alemayehu et al. [19] listed 53 medicinal plants used in treating 22 kinds of livestock diseases in the Amhara region, Northern Ethiopia. Additionally, Luseba and Tshisikhawe [20] outlined the use of 37 medicinal plants used in the treatment of livestock diseases in the Vhembe region, Limpopo Province, South Africa, while Ayeni and Basiri [21] presented a surveyed result of ethnoveterinary plants used in treating livestock diseases among the Fulani people of Girei, Adamawa State of Nigeria. Their survey [21] identified 30 medicinal plants species and the different livestock disease conditions they treat. Dzoyem et al. [6] recorded 138 plants that are used in Cameroon to manage livestock diseases and maintained that an overwhelming majority of animal owners in Cameroon rely on traditional healthcare practices to keep their animals healthy. The current review noted that most plants could be used in more than one animal disease. Considering the veterinary usage of the reported medicinal plants, more pharmacological studies are required for confirming the effectiveness of these herbal remedies.

### 3.2. Preparation of Some Ethnomedicinal Plants and Their Veterinary Uses

Most ethnomedicinal materials noted in sub-Saharan Africa come from trees, herbs, shrubs, and climbers as recorded by Offiah et al. [22], Lawal et al. [23], and Tekle [24]. The frequent usage of trees and shrubs is due to their richness in the forest than other habits [25]. Zorloni [26] reported a decline in the use of woody plants (forest and wood land species) in the Tigray region, Northern Ethiopia. The author further explained that there are more herbaceous plant species as compared to trees and shrubs in that area [26]. Moreover, the trend of using more of herbaceous plants could be advantageous as it is easier to cultivate them when they are in short supply.

Teklay [27] reported grinding or crushing in a wooden or stone mortar and pestle, soaking, or boiling different parts of plants as the common drug extraction methods in the Seharti-Samre district, Northern Ethiopia. Dzoyem [6] noted preparation of roots and barks by boiling, leaves prepared by pounding and soaking in cold or warm water and burning the whole plant or just the useful plant part to ash in Cameroon. Furthermore, Yirga et al. [28] recorded processing of some ethnomedicine by mixing with other ingredients such as water, butter, or coffee. Adding additives such as whey, ghee, sugar, and/or water to ethnomedicines was reported to counteract the astringent taste, dilute, and reduce the relative potency of the remedy [28].

Ethnoveterinary medicine recorded in separate study areas is mainly administered to livestock orally as decoctions; steeped materials used in liquid form; suppositories; through smoke, vapors, massages, or intranasally; or applied topically on the skin or as a bath in skin problems [29,30]. The use of ash from burned medicine through licking or applying it directly on wounds or small cuts was also reported [6,31,32]. Odongo et al. [33] recorded oral dosing of juice made from boiled material to cure internal ailments or systemic treatments, as well as washing sick animals. Tekle [24] stated that most ethnoveterinary practitioners in Zambia prefer administering medicines orally followed by dermal application. Tekle [24] further illustrated that the reason is that oral and topical routes allow rapid physiological reaction with the pathogens, thereby increasing the curative power of the medicines.

The findings of McGaw et al. [34] revealed that ethnoveterinary practitioners use coffee cups, beer bottles, highland plastic, finger length, number of drops, and teaspoons in determining ethnomedicinal dosages. Livestock species, body condition, physiological status, sex, and age are factors considered in the preparation of dosages. As a result, older livestock with good body condition are served with higher amounts and concentrated concoctions than younger and emaciated ones [34]. The lack of standardized dosing observed in most ethnomedicinal applications and administration in Africa leads to skepticism among veterinarians using allopathic veterinary medicines [31].

An ethnobotanical study conducted in Eastern Cape, South Africa, by McGaw and Abdalla [35] confirmed the use of EVM in treating diarrhea, dysentery, and hemorrhage by members of the community. Similarly, Olajuyigbe and Afolayan [36] recorded anthrax, wounds, lymphatic swelling, and bloody urine as livestock ailments treated using EVM. Megersa [37] revealed that EVM are used for the treatment of skin diseases, external parasites such as ticks, helminthiasis, and respiratory disorders characterized by coughing.

### 3.3. Forms and Current Trends of Ethnoveterinary Practices in Sub-Saharan Africa

Ethnoveterinary practices are more common in developing countries due to different socioeconomic factors Teixidor-Toneu and D’Ambrosio [38]. This is especially so in areas of rural sub-Saharan Africa, where livestock diseases are rampant and modern veterinary services are insufficient Katerere and Luseba [39]. Despite the increasing use of modern veterinary services to cater for livestock healthcare needs, traditional remedy remains a prominent complementary medical practice, as recorded in some sub-Saharan African countries such as Angola [4], Botswana [5], Cameroon [6], Ethiopia [7], Kenya [8], Nigeria [9], Namibia [10], South Africa [11], Tanzania [12], Zambia [13], and Zimbabwe [14]. According to Worku [40], this traditional healing comprises belief, knowledge, practices, and skills pertaining to healthcare and management of livestock. According to the World Health Organization, at least 80% of people in developing countries rely mainly on indigenous practices for the control and treatment of numerous diseases affecting both human beings and their animals [41]. Ethnoveterinary practices have retained their popularity in countries of the developing world, and their use is rapidly spreading in sub-Saharan Africa [42].

### 3.4. Threats to Ethnoveterinary Medicine

Ethnoveterinary medicinal materials are rarely stored except for finely crushed materials whose powder is kept and used within a month [43]. Van der Merwe [44] attributed the diminishing ethnoveterinary knowledge to the death of elderly knowledgeable members of society since the documentation of most herbal remedies is handled by the elderly. [31] pointed to the rapid socio-economic, ecological, and technological changes in peoples’ lifestyles as factors leading to the disuse or total loss of traditional knowledge. Odongo et al. [33] further contended that the situation is exacerbated by views that traditional practices are evil, satanic, sinful, and therefore ungodly. The lack of official recognition of the role played by ethnoveterinary practitioners in the prevention, control, and treatment of livestock diseases in some countries is another factor making young people reluctant to maintain traditional morals of the society [33]. In addition, environmental degradation, agricultural expansion, deforestation, and urban development are factors leading to a loss of habitats and ethnobotanical species [45]. Harvesting firewood for charcoal, drought, agriculture, and trade are other factors posing a threat to medicinal plants [45].

### 3.5. Conservation of Ethnomedicinal Plants

Studies by Verma [45], Giday et al. [46], Wabwire [47], and Lulekal et al. [48] have suggested various medicinal plant conservation methods based on threats observed in their study areas. Kadam and Pawar [49] suggested conservation of ethnomedicinal plants in gene banks such as botanical gardens and field gene banks.

Various suggestions include the involvement of governments in conservation measures to reduce the threat on existing ethnomedicinal plants [50]. Abeba [51] recommended awareness campaigns and enhancing of conservation, alongside urgent collection of germplasm, for severely declining species. It is important to raise awareness among youth of the contribution of ethnomedicinal practice toward fulfilling the primary healthcare of livestock. Nefhere [52] proposed the inclusion and incorporation of the usefulness of ethnomedicinal medicine in school curricula to raise the awareness amongst young people and the need for documenting indigenous ethnoveterinary knowledge to ensure conservation of declining medicinal plants. Dzoyem [6] reported that medicinal plant materials are usually not stored but are collected and used fresh when needed; however, if the need arises, they can be stored in powdery form, which is kept in a cool place in plastic or paper bags, newspaper, and glass, metal, or plastic jars away from direct sunlight and wind.

Modernization coupled with poor storage of ethnoveterinary knowledge based on individuals’ remembrance abilities and its transmission from generation to generation by word of mouth has greatly endangered its survival and sustainability into the future [31].

### 3.6. Limitations of Ethnoveterinary Medicine

According to Ezeanya-Esiobu [53], some of the ethnoveterinary remedies are inconvenient to prepare or use because certain plants are available only at certain times of the year. Miglas and Belachew [54] contended that ethnoveterinary medicine dosages are uncertain and not standard as they have an empirical basis. Gabalebatse et al. [5] contended that EVM are difficult to standardize because the pharmaceutical value and concentration of active ingredients in each plant vary depending on climatic and geographical factors. Etana [55] stressed that ethnoveterinary medicines are often not as fast working and potent as conventional medicines. As a result, they may be less suitable for treating epidemic and endemic infectious diseases such as foot-and-mouth disease, rinderpest, anthrax, black quarter, and rabies [55]. However, EVM should not be dismissed out of hand, as many conventional drugs are of plant origin. Ethnomedicinal diagnosis may be inadequate as it is based on symptoms rather than the underlying cause of the disease [26]. The author further argues that a variety of other factors may cause, for example, loss of appetite, restlessness, weight loss, and nervousness—hence the need for proper diagnosis. Kubkomawa [56] and Yirga et al. [28] reported the existence of inappropriate ethnoveterinary practices such as cauterizing the vulva to induce heat, hot iron branding, the use of snake oil (leading to killing or loss of snakes), which have a negative impact on the health of animals and thus need to be discouraged.

## 4. Conclusions

Ethnoveterinary medicine, especially the use of medicinal plants in the treatment of livestock diseases, needs to be scientifically explored. Because sub-Saharan Africa has several medicinal plants, research should be directed toward their potential. The shortage of veterinary drugs and the poor accessibility of modern veterinary healthcare services by rural farmers make the case for the use of ethnomedicinal plants stronger. In addition to being cheaper and more accessible, ethnomedicinal plants can prove to be viable therapeutic options or substitutes if they are properly investigated and standardized. For such effort to materialize, a multi-disciplinary research approach involving veterinary doctors, chemists, botanists, and ethnoveterinary practitioners appears to be the best route to pursue. In addition, strong policy support is required to promote and integrate research, training, and application. While some regulations are in place to monitor the practice of traditional medicine in humans, especially in urban areas, similar regulations for ethnoveterinary care need to be drafted.

## 5. Recommendations

Further study and promotion of ethnoveterinary medicine are bound to help communities conserve information and integrate select practices into rural animal healthcare services. This may create opportunities for phyto-chemists and pharmacologists to conduct future studies. Two points are worth highlighting. First, local knowledge can be converted into medicinal or other commercial products. Local people and the keepers of this knowledge should be recognized and appropriately compensated. Second, over-exploitation of medicinal plants is bound to put their survival at risk, and measures need to be implemented to conserve them.

## Data Availability

Not applicable.

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
