# Peer review of "Review on Ethnoveterinary Practices in Sub-Saharan Africa"

_vetsci, 2021, doi:10.3390/vetsci8060099_

Round 1
Reviewer 1 Report
The review article entitled “Review on ethnoveterinary practices in Sub-Saharan Africa” is a manuscript that falls within the scope of the journal Veterinary Sciences. It can be published in this international journal, but with a thorough revision regarding: (1) the reduction and rewrite of the Background, (2) the inclusion of a comment section about the most used plant species, and (3) the correct presentation of the references, both in the text and in the final list.
Within the Results section, the sub-section titled “Definitions and origin of Ethnoveterinary Medicine” does not present results, this is just a long series of comments related to the EVM concept and its origin. All of this should move on to the introduction of the review.
Table 1 is not cited in the text or commented on a specific section. As a reader it is not clear to me what the content of this brief table is. Are these the plant species used in Sub-Saharan EVM? … No, obviously it cannot be only 23 species. If these are the plants for which studies have been carried out on their veterinary effectiveness, commenting on their properties and recommended uses is very important.
In relation to this table I also want to comment on another important thing. To avoid ambiguities and errors, pharmacological or biomedical studies on plants require correct use of botanical scientific nomenclature. Correct spellings, accepted names, author citations and current family designations are crucial (see Bennett and Balick, 2014; Rivera et al., 2014) [Bennett, B.C., Balick, M.J., 2014. Does the name really matter? The importance of botanical nomenclature and plant taxonomy in biomedical research. J. Ethnopharmacol. 152 (3): 387-392 / Rivera, D., Allkin, R., Obón, C., Alcaraz, F., Verpoorte, R., Heinrich, M., 2014. What is in a name? The need for accurate scientific nomenclature for plants. J. Ethnopharmacol. 152 (3): 393-402]. In order to properly use botanical nomenclature check The Plant List (www.theplantlist.org/)... For example, writing “Vernoni amygdalina Del. (Asteraceae)” is incorrect… the scientific name, authorship and botanical family of this plant species are Vernonia amygdalina Delile (Compositae). Review the taxonomical data of all plant species mentioned in the review is necessary.
In the final list (pages 7-10) the references are arranged in alphabetical order. This is a mistake. The first reference that can be found in the text of the article (page 1 - line 33) is number 69. Please, consult the link “Reference List and Citations Style Guide for MDPI Journals” (https://www.mdpi.com/authors/references).
Reference citation numbers are correctly placed in square brackets, but the name of the first author is not indicated, as requested. For example:
Page 2 / line 50 … Furthermore, according to [5] the delivery of… / Furthermore, according to Ahuja et al. [5] the delivery of…
Page 2 / line 54 … [36] further argued that… / Katunguka-Rwakishaya et al. [36] further argued that…
Page 2 / line 69 … Moreover, study by [3] listed… / Moreover, study by Adedeji [3] listed…
Review the format of all references, according to the Citations Style Guide for MDPI Journals.
Other specific comments…
Page 2 / line 68 … Despite its benefits, ethnoveterinary practice…
Page 2 / line 96 … progress in EVM and to create an understanding of this important field by…
Page 6 / line 191 … a month [31]. [Error! Reference source not found.] ??????
Page 6 / line 218 … in–situ … in situ
Author Response
Thank you for your kind review. Please see the attachment.

Reviewer 2 Report
The review deals with ethnoveterinary drugs and remedies used in sub-Saharan Africa. The topic is interesting, even beyond its strictly medical value, because of its cultural, social, and economic implications. The bibliographic research looks accurate, but please note that references should be listed in order of appearance in the manuscript.
The information looks complete, but, in my opinion, a section about the effectiveness of some EVM practices would greatly enrich the paper.
Language needs revision since it is quite good, but many little mistakes need to be fixed. Following there are just some examples:
Line 34: Please use “result” instead of “results”.
Line 35: Please use “impact” instead of “impacts”
Line 36: Please enclose “although relying on livestock” between commas.
Line 40: Please do not use a comma between subject and verb.
Further linguistic problems are not yet reported.
Following there are some specific comments.
Lines 46-47: Please check this sentence.
Line 102: Please use “systematic” instead of “Systemic”.
Line 162: Please explain why vaccination is considered an administration route.
Lines 190-195. This paragraph should be moved to the next section (Conservation of ethnomedicinal plants).
Lines 191-195. Please check the reference.
Lines 221-229: This paragraph should be moved to the previous section (Threats to ethnoveterinary medicine)
Author Response

(The authors gave the same response as above.)

Reviewer 3 Report
Dear authors,
Thank you for your interesting work. One of the major changes I would like you to do is to replace all the reference numbers in the text with the names of the authors. I also recommend to re-write the manuscript as it needs clarification and all the information provided to be contributed in relevant appropriate paragraphs.
Additionally, I would like to see:
- a separate paragraph with the main forms of ethnoveterinary practices in sub-Saharan Africa
- Currents trends ethnoveterinary practices and how they are affected
- Provide numbers or percentages of how many people are estimated to use these practices
Author Response

(The authors gave the same response as above.)

Round 2
Reviewer 1 Report
No comments.
Reviewer 3 Report
Dear authors,
Thank you for your prompt response. I am satisfied with the corrections and pleased to accept the review in the current form.